# Multiple Long-Term Conditions (MLTC) and the Environment: A Scoping Review

**DOI:** 10.3390/ijerph191811492

**Published:** 2022-09-13

**Authors:** Hajira Dambha-Miller, Sukhmani Cheema, Nile Saunders, Glenn Simpson

**Affiliations:** 1Primary Care Research Centre, University of Southampton, Southampton SO16 5ST, UK; 2Swansea Medical School, University of Swansea, Swansea SA2 8PP, UK

**Keywords:** MLTC, multimorbidity, environmental determinants

## Abstract

**Background:** Multiple Long Term conditions (MLTC) are a major health care challenge associated with high service utilisation and expenditure. Once established, the trajectory to an increased number and severity of conditions, hospital admission, increased social care need and mortality is multifactorial. The role of wider environmental determinants in the MLTC sequelae is unclear. **Aim:** the aim of this review was to summarise and collate existing evidence on environmental determinants on established MLTC. **Methods:** comprehensive search of Medline, Embase, Cochrane, CINAHL and Bielefeld Academic Search Engine (BASE), from inception to 4th June 2022 in addition to grey literature. Two authors independently screened and extracted papers. Disagreements were resolved with a third author. **Results:** searches yielded 9079 articles, 12 of which met the review’s inclusion criteria. Evidence of correlations between some environmental determinants and increased or decreased risks of MLTC were found, including the quality of internal housing/living environments, exposure to airborne environmental hazards and a beneficial association with socially cohesive, accessible and greener neighbourhood environments. **Conclusions:** The majority of the 12 included papers focused on the built and social environments. The review uncovered very limited evidence, indicating a need for further research to understand the role of environmental determinants in MLTC.

## 1. Introduction

Multiple Long Term conditions (MLTC), which is often used interchangeably with the term multimorbidity, are a rapidly growing health care challenge due to increased longevity and a rising prevalence of chronic health conditions. The Academy of Medical Sciences define MLTC as: ‘The coexistence of two or more chronic conditions, each one of which is either a physical noncommunicable disease of long duration, such as a cardiovascular disease or cancer; a mental health condition of long duration, such as mood disorder or dementia; an infectious disease of long duration, such as HIV or hepatitis C’. [1] (pp. 22–23). Where two or more health conditions coexist in this way, medical and social care needs are greater. This results in higher levels of service utilisation and costs [2] to a wide range of health and social care providers [3,4], for example, those living with MLTC have higher rates of unplanned hospital admission, longer durations of hospitalisation, require higher rates of polypharmacy and are more likely to be affected by adverse drug reactions [5,6].

The prevalence of MLTC will grow significantly over the next decade, with the number of people aged 65-plus in England with two or more diseases increasing from 54% in 2015 to a projected 67.8% by 2035 [7]. Furthermore, it is estimated that by 2035 around 17% of the UK’s population will have four or more chronic conditions, nearly double the existing prevalence of complex MLTC [8]. Whilst MLTC is more prevalent with age, it is not limited to older age groups, with MLTC also being more common within other population cohorts such as women and ‘those from low socioeconomic backgrounds, particularly in high-income countries’ [9] (p. 2).

The projected growth in MLTC poses a significant challenge ‘to current healthcare systems that are designed to treat and manage single health conditions’. [10] (p. 1). As Guthrie et al. 2012 [11] (p. 1) comment, managing MLTC is challenging as: ‘Clinical decision making is more difficult in people with multimorbidity because clinicians and patients often struggle to balance the benefits and risks of multiple recommended treatments and because patient preference rightly influences the application of clinical and economic evidence’. There are also substantial health and social costs for individuals with MLTC, who are more likely to experience severe illness and complication rates, increased physical and mental disability, lower quality of life and risk of social vulnerability including homelessness, unemployment and poverty [12,13,14].

A growing body of evidence has highlighted the multifactorial causes of MLTC including wider health, social and environmental determinants [15]. Environmental determinants are those that occur in natural, built or social environments [16]. The built environment refers to “the characteristics (objective and subjective) of a physical environment in which people live, work and play, including schools, workplaces, homes, communities, parks/recreation areas, green (i.e., visible grass, trees and other vegetation) and blue spaces (i.e., visible water)” [17] (p. 6). The quality of the natural and built environment can have either a deleterious or beneficial impact on the health of an individual or the wider population health of a community [18]. Much of the literature focuses on the harmful effects or health risks associated with poor quality natural and built environments [1]. Understood in this context, environmental determinants of health are those attributed to “exposure to pollution and chemicals (e.g., air, water, soil, products), physical exposures (e.g., noise, radiation), the built environment (e.g., housing, land-use, infrastructure), other anthropogenic changes (e.g., climate change, vector breeding places), related behaviours and the work environment” [19] (p. 465). As Pruss-Ustin (2017) [19] comment, these human-made or modified determinants of health that are external to the person, can be ameliorated or prevented by policy interventions “either with almost immediate effect, or with longer term transformations” (p. 465). Interventions to address environmental determinants in relation to the built and physical environment typically focus on a range of policy areas, which directly or indirectly affect health and wellbeing of individuals and populations [20]. ‘Direct’ environmental determinants are those ‘traditionally associated with infrastructure planning and environmental health’ (e.g., air quality, noise pollution and traffic-related factors) and where health impacts are ‘quantifiable and causal effects can be attributed’ [21] (p. 4). Indirect determinants, which can be more difficult to measure are: “the ways in which built environment features and their design can influence the feelings and behaviour of individuals and populations. For example, perceptions of the local area, social connections, accessibility and physical activity levels” [21] (p. 8). These forms of indirect determinants are associated with social determinants of health [22]. Importantly, built and physical environment determinants of health are often spatially concentrated (i.e., spatial inequalities in health), with deprived areas, populations and communities being more disproportionately affected as a result of living in degraded and/or poorly designed physical spaces and environments [18].

In MLTC, there is limited evidence on the role of environmental determinants, especially in relation to the trajectories of those with established MLTC towards declining health, mortality and increased care needs. The limited studies that have been published in this field have tended to focus on biological or clinical determinants such as weight changes or variations in blood markers [19,22]. This is owing to the fact that these determinants are more easily available and relatively straightforward to measure, compared to environmental determinants.

Understanding an individual’s MLTC trajectory and subsequent outcomes in relation to their immediate environments could provide opportunities to identify key events in their life course at different geographical scales. In turn, this could help identify populations at risk of worsening MLTC trajectories by localisation (e.g., within a neighbourhood context) and exposures within their surrounding environment, alongside health and social determinants. This is essential in designing effective solutions and influencing response mechanisms within health services as it can indicate locations in need of interventions and allocation of available resources. Accordingly, in this review we collate and summarise evidence on environmental determinants that have been examined in relation to established MLTC.

## 2. Materials and Methods

### 2.1. Review Approach

The study followed the Preferred Reporting Items for Systematic Reviews and Meta-Analyses (PRISMA) guidelines for scoping reviews [23]. The scoping review method allowed for a rapid mapping and assessment of key existing research and emerging evidence in this field [24].

### 2.2. Search Strategy

Systematic electronic searches were conducted from database inception to 4 June 2022 on Medline, EMBASE, The Cochrane Library, CINAHL and Bielefeld Academic Search Engine (BASE). For searches of electronic databases, free-text and MeSH terms Cwere used in relation to ‘multimorbidity’ and ‘environmental’. Detailed search terms are shown in Table 1.

### 2.3. Inclusion/Exclusion Criteria

Articles were eligible for inclusion if published in the English language, were related to adults aged over 18 years with MLTC and included mention of any environmental determinants relevant to the review’s aims.

Quality assessment criteria are not a priority for scoping reviews [18], therefore extracted articles were not excluded on this basis.

### 2.4. Study Selection and Data Extraction

All articles identified were imported into the Rayyan collaborative systematic review platform for screening, which was conducted in blinding mode. Rayyan enabled rapid screening of retrieved sources. Titles and abstracts were screened, with each article assessed for relevance according to the inclusion criteria.

Full-text versions of potentially relevant articles identified during the initial screening process were retrieved for detailed assessment. Both screening and data extraction of the full-text articles were conducted independently by two reviewers and disagreement resolved by discussion. A data charting form was used to record and collate the studies and identify a range of ‘key characteristics’ (Appendix A). Reviewers extracted the article reference and date, the stated aim of the study, methods, results, variables and indicators of environmental determinants, and the key findings. Final screening of the most relevant articles was conducted and any disagreement between reviewers was resolved with a third reviewer.

### 2.5. Summarising and Analysis

The data gathered by the review were iteratively synthesised descriptively, using counts to summarise article characteristics (that set out the number, type and quality of studies extracted); use of the data charting technique; and through interpretation of the findings by sifting and sorting material [25]. The main environmental determinants identified by the review were collated and presented below.

## 3. Results

### 3.1. Screening Process

In total, searches identified 9079 articles, which were placed on the Rayyan systematic review tool for screening. Following title and abstract screening and removal of duplicates using the Rayyan, 138 articles underwent full-text screening, which resulted in a further 126 articles excluded from the review. Reasons for exclusion were:-the article did not analyse environmental determinants;-the article did not specifically focus on the topic of multiple long-term conditions/multimorbidity;-the article was not related to the study population of adults aged 18+.

A final total of 12 articles were included in the review. A flowchart (Figure 1) illustrating the screening process, including the reasons for exclusion, is shown in the Appendix A.

### 3.2. Characteristics of Included Studies

Studies were located in a variety of countries and international regions, most frequently the USA (n = 3), followed by Australia and Canada (n = 2), and UK, South Africa, Sub-Saharan Africa, North America-Europe-Australasia and no specified geography (n = 1). The main study settings identified were community/neighbourhood (n = 6), community and people at home (n = 3), no specific setting (n = 2) and low-income housing developments (n = 1). The main age category found in the studies was 18 years and over (n = 7), with the remaining studies focused on middle and older aged populations of 45 years and above, while one study did not specify a specific age range. A wide variation in the sample size was found among the studies, ranging from a sample of 20 (in a mixed-methods study) to a population of 408,111 (in a quantitative study). The main study type was quantitative (n = 6), followed by both mixed methods (n = 3) and reviews (n = 3).

The characteristics of the included articles are summarised in Appendix A.

### 3.3. Summary of Environmental Factors

Studies discussed a range of environmental determinants and possible associations with MLTC, which included those relating to the natural, built and/or social environments. These are discussed below (more details on each numbered study can be found in Appendix A).

#### 3.3.1. Built Urban Environment

Five studies considered various aspects relating to the effect of urban environments and neighbourhood characteristics on MLTC. One study (I) examined environmental determinants which supported the ‘maintenance of functional ability’. This study highlighted the importance of physical proximity in urban contexts for the cohort aged 85 and over with MLTC, concluding that ‘proximity to local grocery shops may support their functional ability’ to undertake grocery shopping ‘over a five-year period’. A significant limitation of the narrow analytical focus of this study was that it did not consider the impact of how ‘other environmental factors (safety, traffic, stairs, the quality of pavement, age-friendliness of local shops) and individual factors (poverty, socioeconomic disadvantage) might affect use of local shops’.

Another study (II) considered whether the built and urban environments, including its condition, configuration and layout inhibited or facilitated walking (‘walkability’), along with accessibility to recreation parks in urban neighbourhoods. Findings indicated that ‘living in a walkable neighbourhood and having higher park accessibility is associated with lower odds of hypertension, especially for lower income individuals’. The authors suggested the need for ‘an integrated population health approach that considers multimorbidity as a result of exposure to car-dependent areas and the lack of green spaces’.

A study examined (III) exposure to outdoor air pollution and/or hazards relating to air quality and associations with the development of chronic conditions in low to medium income countries (LMICs). The results indicated that compared with developed states, populations in LMICs are potentially more exposed to ‘poorer air quality and increased risk of multimorbidity’ because of factors including ‘overpopulation and rapid urbanisation coupled with developing industrialisation’, as well as higher usage of and reliance on ‘wood fuel’ due to higher levels of poverty. This study also highlighted evidence of a link ‘between long term indoor exposure to air pollution and cancer in LMICs, and CVD [Cardiovascular disease] and related mortalities’.

One study (IV) examined conditions within household environments, in particular the effect of ‘housing-related environmental exposures’ on health in low-income households. Using ‘binary indexes and a summed index’ covering six household exposures (these were: mould, combustion by-products, second-hand smoke, chemicals, pests, and inadequate ventilation), the study identified ‘significant clustering of effects in [the] housing site for 4 of the 6 indexes: pests, combustion by-products, mould, and ventilation’. The study found that household ‘environmental problems were common’ with ‘more than half of homes had 3 or more exposure-related problems’ and these can be associated with asthma development and respiratory conditions. A major weakness reported was that the ‘study design was cross-sectional and thus could not determine the causal relationship between exposures and health,’ including any association with MLTC.

Another study (V) sought to determine ‘evidence gaps to building capacity in supportive housing health care research about chronic diseases and multimorbidity, identify ongoing strains and offer potential solutions’. It presented a ‘multifaceted framework’ of eight priorities for action, including ‘scaling up environmental determinants to cover residential spaces’; implementing acute methodologies for indoor environment risk assessment;’ and ‘increasing clinical research applicability into built environment studies’. The authors recommended that future policy interventions should explore ‘approaches to strengthen health and care delivery through housing design solutions for people living with chronic morbidity’.

#### 3.3.2. The Built and Social Environments

Two studies examined both the built and social environments. One study (VI) explored the ‘association between neighbourhood characteristics [defined as five ‘contextual neighbourhood factors’ of: neighbourhood-level crime, accessibility to health care services, availability of green spaces, neighbourhood obesity, and fast-food availability] and type 2 diabetes (T2D) comorbidity in serious mental illness (SMI). The results showed that individuals with SMI living in high crime areas ‘had 2.5 times increased odds of reporting T2D comorbidity’ compared with those in ‘lower crime rate areas’. No causal association was found in relation to the other four neighbourhood characteristics variables.

Another study (VII) explored self-rated health and its association with multiple types of perceived environmental hazards, including participant’s perceptions of neighbourhood ‘environmental hazards (e.g., air quality, odours and noise)’, self-reported ‘aspects of the social environment (e.g., feeling safe, neighbourhood crime, social cohesion)’ as well as ‘culture-related stressors (e.g., immigration status, language stress, ethnic identity)’. The findings highlighted ‘negative perceptions of environmental hazards and reported cultural stressors were significantly associated with fair/poor self-rated health among residents in a low-income majority-minority community’. Social cohesion was found to have a ‘beneficial association with self-rated health’.

#### 3.3.3. The Social Environment

The social environment was considered by four studies. Housing and wider living conditions were the focus of one study (VIII), which examined ‘social and structural barriers’ among those with co-occurring disorders (COD), that is ‘mental illness, substance abuse, and general medical conditions encounter[ed] in regard to their health care’. Social barriers were experienced by those with COD including difficulties relating to their interpersonal ‘relationships with health care providers’ and ‘negotiating an arduous the healthcare system’ (e.g., problems relating to ‘inadequate insurance coverage and medication capitations’). These challenges were compounded by wider environmental determinants, in particular ‘living in contexts that made health management difficult’. These included having to live in ‘unstable’ multi-occupancy accommodation with other individuals who had mental health and substance misuse problems, living in poor quality accommodation with rodent and insect infestations and residing in areas with high crime rates. 

Another social environment study (IX) focused on identifying household and area-level social determinants of multimorbidity and co-morbidity. Across three main social determinants of household composition, household tenure and household rurality, mixed evidence was found of these determinants being associated with either increased or decreased multimorbidity. Much of the evidence presented in this study was found to be inconclusive in relation to the role and effect of household and area-level social determinants on multimorbidity.

The third study (X) aimed to ‘determine the prevalence of multimorbidity and examine its association with various social determinants of health in South Africa’. Variables including age, gender, education, income, employment, obesity, depression, income and smoking, were found to be associated with multimorbidity. The authors also reported that social capital (defined as ‘how people connect with others in their environment’), which has been shown in the literature to be beneficial for health, was not associated with multimorbidity or even health per se.

Another study (XI) focused on identifying ‘associations between lifestyle behavioural factors and multimorbidity resilience (MR) among older adults’. A broad range of variables were used including four social and environmental variables: number of friends, number of relatives, housing problems and urban/rural status. The results indicated all four factors ‘exhibited statistically significant, albeit weak, associations with multimorbidity’.

#### 3.3.4. The Natural Environment

The natural environment was the focus of one study (XII). This explored the effects of environmental degradation (resulting from dry salinity) on mental health in relation to ‘diseases co-morbid [asthma, ischaemic heart disease, suicide/self-inflicted injury] with depression in this environmental setting’. It found that ‘the association of asthma, suicide and heart disease with salinity was most likely attributable to the co-morbidity of the conditions with depression’.

## 4. Discussion

This rapid scoping review was conducted to identify and gain a broad overview [19] of the key research evidence on the role of wider environmental determinants in the MLTC sequelae. All but one study identified by this review considered either the built environment or social environments or both, and how these aspects of environmental determinants related to MLTC. While a relatively broad range of specific determinants were examined in the studies identified by this review, there was an analytical focus on various aspects of the quality of housing/accommodation and living conditions, external air pollution and indoor air quality, and the design and configuration of urban environments and neighbourhood characteristics (e.g., neighbourhood levels of crime, accessibility to healthcare services, availability of green spaces).

These studies did find evidence of correlations between these determinants and increased or decreased risks of MLTC, particularly in relation to the quality of internal housing/living environments, exposure to airborne environmental hazards (e.g., air pollution/air quality). Further, there was some evidence that socially cohesive, accessible and greener neighbourhood environments had a beneficial association with health, although it was not clear to what extent this relates to MLTC.

It is not clear why these aspects of environmental determinants featured more prominently in this scoping review, although one reason might be that they are more easily quantified and measured, and data sources (including electronic health, public health and environmental health records) are more readily available and accessible to researchers. Additionally, these environmental determinants may also be more prevalent and widely reported, and as such there is wider societal awareness of these determinants, which in turn may have led to greater scrutiny and examination of possible associations with MLTC.

Very little evidence was found on the role of the natural environment on established MLTC, suggesting this aspect of environmental determinants is a particularly under-researched field.

## 5. Strengths and Limitations

We carried out a rapid scoping review to summarise evidence on MLTC and environment that could inform further research in this field. We included broad inclusion criteria of MLTC and any environmental determinants to help identify a wide range of studies including grey literature.

The main limitation of this scoping review is the small number of relevant sources that were identified, which limits comparison with the findings of previous studies. Another limitation of this study is that more specific search terms around MLTC and environmental determinants, including, for example, housing, pollution and air quality, etc; may have increased the number of papers identified. Further, we restricted papers to those published in the English language, which could have limited identification of all relevant papers in this area. We did not carry out quality assessment of papers, as this is not a requirement of scoping reviews.

## 6. Conclusions

This scoping review aimed to collate evidence on environmental determinants in MLTC. We identified only 12 relevant papers in this field, and most of these focused on the built and social environments. The evidence on the natural environment was extremely limited. Overall, this review uncovered very limited evidence on environmental determinants of MLTC. Indeed, much of the available evidence did not establish a causal relationship between specific exposures and health or only found weak associations. This finding makes it difficult to reach any definitive conclusions on this subject. It is clear however, that there is a particular gap in the literature relating to the natural environment and possible effects on established MLTC. Further, there is a paucity of evidence in relation to which combinations of environmental determinants may have beneficial or deleterious effects on an individual’s MLTC trajectory and subsequent outcomes across the life course. Our study indicates that significant further work, perhaps in the form of an in-depth systematic review, is required to begin to address these extensive gaps in the evidence base.

## Figures and Tables

**Figure 1 ijerph-19-11492-f001:**
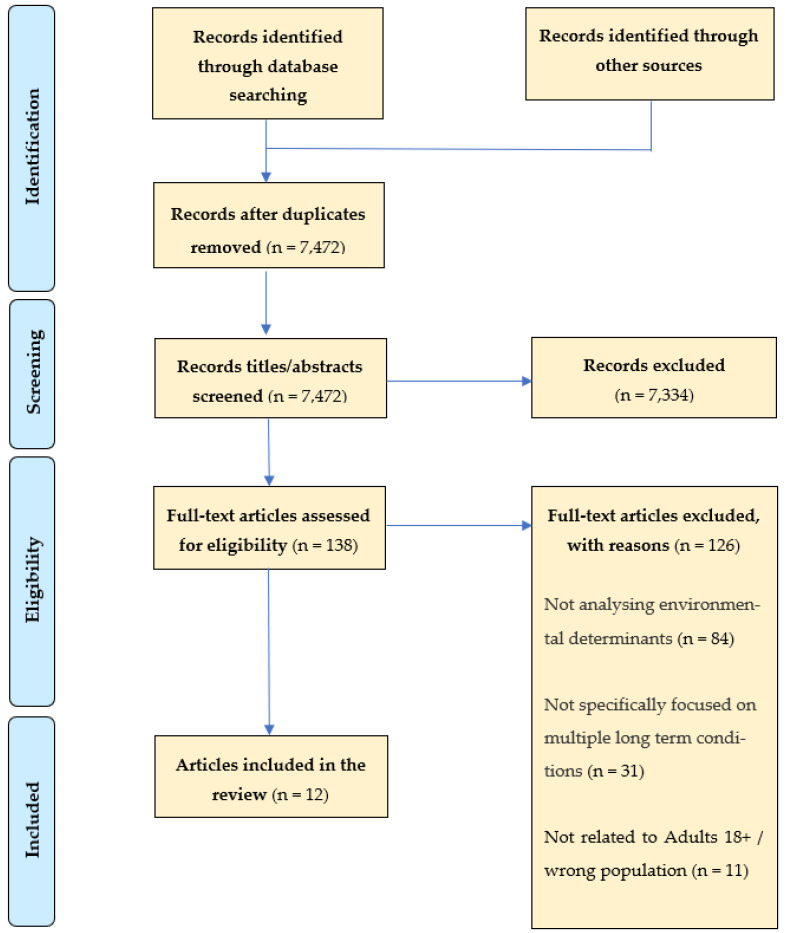
Adapted PRISMA Flow Chart. Explaining the study’s documentary screening and inclusion/exclusion process.

**Table 1 ijerph-19-11492-t001:** Search terms results table.

MeSH and Free Text Search Terms	Databases Searched	Filters/Refined by	Number of Sources Identified
(1) MESH: Multimorbidity (or equivalent on database) FREE TEXT: MLTC OR MLT-C OR MLTC-M OR MLTM OR Multiple Long Term Conditions OR Multimorbidity OR Multimorbid OR Multimorbidities OR Co-morbid OR Multiple Conditions (2) MESH: Environmental (or equivalent on your database) FREE TEXT: Green OR open space OR Housing OR Mobility OR Air pollution OR Air Quality OR noise OR water quality	Medline	All dates searched. Language: restricted to English/English Language.	550
	Embase	All dates searched. Language: restricted to English.	1031
	Cochrane Library	All dates searched. Language: restricted to English.	5092
	Cumulative Index to Nursing and Allied Health Literature (CINAHL)	All dates searched. Language: restricted to English.	2173
‘MULTIMORBIDITY’ AND ‘ENVIRONMENTAL DETERMINANTS’ doctype:1 (free text AND mesh terms specific to BASE database) = 98 MULTIPLE LONG TERM CONDITIONS AND ‘ENVIRONMENTAL DETERMINANTS’ doctype:1 (free text AND mesh terms specific to BASE database) = 129	BASE	All dates searched. Language: restricted to English. Entire document.	227
	Manual searches/references/ expert input		6
			Total = 9079

Manual searching of bibliographies were also conducted. The views of topic experts were also sought to identify possible additional sources.

## Data Availability

Data used during the current study are available from the corresponding author on reasonable request.

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
