# Peer review of "Multiple Long-Term Conditions (MLTC) and the Environment: A Scoping Review"

_ijerph, 2022, doi:10.3390/ijerph191811492_

Round 1
Reviewer 1 Report
I'm glad to have the opportunity to review the manuscript. It raises an important topic: environmental impact on MLTC.
The presented data are very interesting, but the manuscript needs improvement.
Below, I pointed out my comments:
1. Specify the title by indicating in what context the environment is related to MLTC.
2. Is citation 17 correct? The Prisma for scoping review is described in the article: DOI: 10.7326 / M18-0850.
3. The literature is very poor. Although this is a scoping review, it is worth using more comprehensive sources in the discussion or the introduction.
4. In the results, the numbering of publications included in the review is according to the appendix (for example, line 219). This is not clearly presented. It is better to move the appendix to the publication's content or add information at the beginning of the results, what this numbering refers to.
5. In conclusion, it is necessary to indicate what gaps have been identified. Please show which areas can be prepared for a systematic review based on more specific search criteria.
Reviewer 2 Report
This is a nicely written concise review of selected published articles about multiple long-term conditions (MLTC) and the environment. Twelve articles were included in the review, and some of the findings showed correlations between environmental determinants and increased or decreased risks of MLTC.
A few comments are listed below for author consideration in order to further improve the quality of the review.
1. Abstract: In line 10, correct "Multiple Long Term conditions (MLTC) is a major health care challenge..." to "Multiple Long Term conditions (MLTC) are a major health care challenge..." Also, in line 21, complete the sentence "The search yielded 9,079 articles of which 12 were included." Clarify where the 12 articles were included (in the review)? In line 26, correct the statement " Only 12 relevant papers were identified..." From the texts of the review 9079 articles were identified but only 12 met the review's inclusion criteria.
2. Introduction: More about MLTC should be included here. How is MLTC defined? What are the criteria for concluding that an individual has MLTC? What is the difference between terms MLTC and multi-morbidities?
3. Study selection and data extraction: Delete the repeated sentence in lines 123-125, "Titles and abstracts were screened, with each article assessed for relevance according to the inclusion criteria."
4. Table in Appendix A is too big and should be formatted using a smaller font.
Best of luck!
Round 2
Reviewer 1 Report
Thank you for submitting your improved manuscript. All suggestion has been made. I have no additional comments.